# BONE: BLOCK-AFFINE ADAPTATION OF LARGE LANGUAGE MODELS

## ABSTRACT

Low-Rank Adaptation (LoRA) has achieved remarkable training results by freezing the original weights and training only low-rank matrices, establishing itself as the predominant fine-tuning method for LLMs. Many LoRA variants have emerged, yet they lack a design tailored to the characteristics of LLM weights and fail to leverage the original weights effectively. To address the sparsity of LLM weights, and drawing inspiration from GQA and MQA, we propose Block-Affine Adaptation (Bone), a novel PEFT technique distinct from LoRA. By dividing the original weights into multiple subspaces that share a single matrix for weight updates, Bone simplifies the process by requiring the trainable matrix to be initialized to zero, eliminating the need for complex initialization as in some LoRA variants. Compared to LoRA, Bone significantly reduces memory usage and achieves faster computation. Evaluation of both NLU and NLG tasks demonstrates that Bone substantially outperforms LoRA and its variants. Inspired by Pissa, we propose a new theory called "Weight Guide" to better utilize the information embedded in the original weights. This approach extracts valuable information through a linear transformation of the original weight matrix using a trainable matrix. To validate the effectiveness of "Weight Guide" we combined it with Bone to create a new structure called Block-Affine Transformation (Bat), and ablation experiments confirmed the effectiveness of "Weight Guide".

## 1 INTRODUCTION

Large models have been integrated into various industries, revolutionizing many traditional technologies Radford et al. (2019); Raffel et al. (2020). However, general-purpose large models often struggle to meet the needs of all downstream tasks, making it necessary to fine-tune base models for specific scenarios. Full-scale fine-tuning of large models is computationally costly; for example, finetuning the LLaMA2-7B Touvron et al. (2023) model with bfloat16 Wang & Kanwar (2019) precision requires around 60GB of VRAM. In contrast, PEFT (Parameter-Efficient Fine-Tuning)Xu et al. (2023) techniques could reduce the VRAM requirement to fit into a 24GB VRAM GPU. As a result, numerous PEFT techniques and quantization methods have emerged to reduce the training costs of large models. LoRA (Low-Rank Adaptation) Hu et al. (2021) has become one of the most popular PEFT methods due to its small tunable parameter size, its effectiveness, and the possibility of zero inference overhead after finetuning.

LoRA significantly reduces memory usage by freezing the original weights $W$ and updating two low-rank matrices $A$ and $B$. Typically, either $A$ or $B$ is initialized to zero, ensuring that the initial state of LoRA is consistent with the pre-trained model, The figure 2 illustrates the structure visualization. However, extensive experiments Ding et al. (2023); Liu et al. (2024b); Biderman et al. (2024) have shown that LoRA's convergence is significantly slower compared to full fine-tuning. This slow convergence is likely due to the small gradients caused by the zero initialization of either $A$ or $B$. To address this issue, researchers have proposed several LoRA variants, such as LoRA+ Hayou et al. (2024), PISSA Meng et al. (2024), and LoRA-GAWang et al. (2024). Despite their excellent performance, these LoRA variants inevitably introduce complexity into the fine-tuning process and reduce structural flexibility. For example, LoRA+ requires manual adjustment of different learning rates for $A$ and $B$; PISSA necessitates performing SVD decomposition on $W$ at the beginning of training, which can be time-consuming when the model parameters are large. As research on LoRA

becomes increasingly saturated, it is essential to explore new directions for PEFT techniques and effectively leverage the unique characteristics of LLMs.

In this work, we first designed Block-Affine Adaptation (Bone), which divides the original weights into multiple subspaces that share a single trainable matrix for updates, making it more efficient than LoRA's two low-rank matrices. Secondly, to address the limitations of LoRA and fully utilize the information in the original weights, we propose a new theory, "Weight Guide". "Weight Guide" enables feature extraction by applying a simple linear transformation to the original weights. Finally, to validate the effectiveness of "Weight Guide", we combined Bone with "Weight Guide" to develop Block-Affine Transformation (Bat).

Our extensive evaluation shows that Bone not only excels in both Natural Language Understanding (NLU) and Natural Language Generation (NLG) tasks but also significantly outperforms LoRA and its variants. Additionally, Bone retains the advantages of LoRA, such as ease of use and no additional computational overhead for the model. In NLG tasks, Bone demonstrates superior performance, with evaluation metrics surpassing even the strong LoRA variant Pissa across the board. To verify the feasibility of Bone, we conducted experiments on two different LLM architectures (LLaMA2 Xu et al. (2023), RWKV6 Peng et al. (2024)). As shown in Figure 1, Bone achieves the fastest convergence, and the results on the test set demonstrate its superior data fitting and generalization capabilities ( Table 2). As a completely new structure distinct from LoRA, Bone offers clear improvements in computational efficiency and memory savings, as detailed in table 7.

**Our contributions can be summarized as follows**

**1.** We propose a novel PEFT technique called Block-Affine Adaptation (Bone). Bone outperforms LoRA and its variants across various fine-tuning tasks. Additionally, Bone is more memory-efficient and computationally faster compared to LoRA.

**2.** To effectively leverage the implicit information in the original weights, we propose the "Weight Guide" theory. By integrating this theory with Bone, we design Block-Affine Transformation (Bat), demonstrating the effectiveness of "Weight Guide".

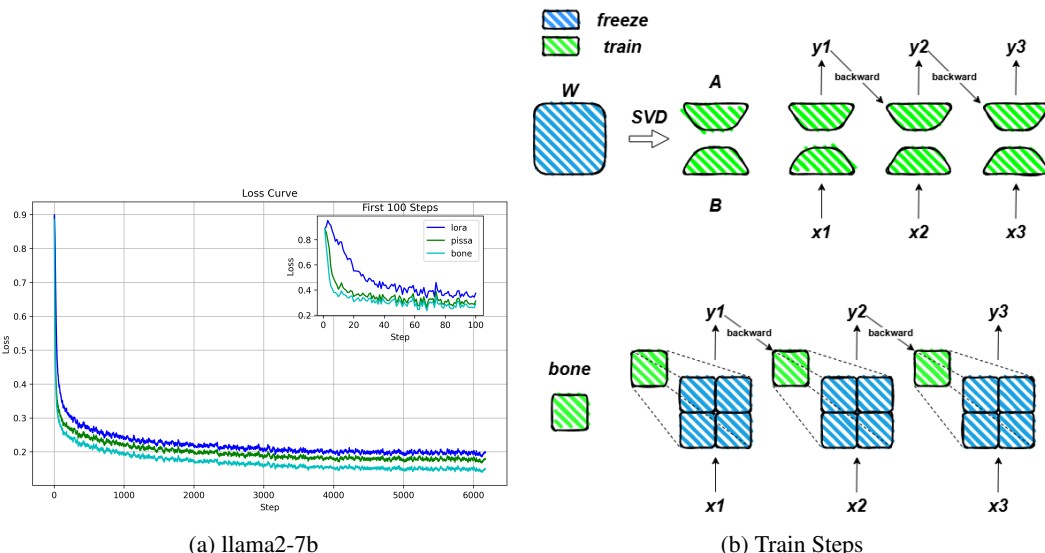

(a) llama2-7b         (b) Train Steps

Figure 1: The left image shows the loss curve for LLaMA2-7B fine-tuned on the MetaMathQA dataset, with the first 100 steps highlighted for closer observation. Comparing the loss curves reveals that Bone demonstrates superior fitting ability across various architectures and parameter settings. Additionally, Bone exhibits a rapid decrease in loss within the first 100 steps, highlighting its effectiveness. The left image is a demonstration of the training state.

## 2 RELATED WORKS

The PEFT (Parameter-Efficient Fine-Tuning) techniques are diverse and include approaches like adapter tuningHoulsby et al. (2019); He et al. (2022); Wang et al. (2022); Pfeiffer et al. (2020), prefix tuningLiu et al. (2023); Li & Liang (2021), prompt tuningBrown (2020); Liu et al. (2023); Lester et al. (2021); Razdaibiedina et al. (2023); Li & Liang (2021), LoRAHu et al. (2021); Meng et al. (2024); Wang et al. (2024); Si et al. (2024), and layer-freezing methods such as LISA.

The adapter method does not require fine-tuning all the parameters of the pre-trained model. Instead, it introduces a small number of task-specific parameters to store knowledge related to that task, thereby reducing the computational demands of model fine-tuning. Prefix tuning is a lightweight fine-tuning method for generative tasks. It adds a continuous, task-specific vector sequence, called a prefix, to the input. Unlike prompts, prefixes are entirely composed of free parameters and do not correspond to actual tokens. Compared to traditional fine-tuning, prefix tuning only optimizes a set of prefixes related to specific tasks, without any change to the original model.

Prompt tuning defines a unique prompt for each task, prepending it to the input data while freezing the pre-trained model during training. LoRA allows us to indirectly train certain dense layers in a neural network by optimizing low-rank decomposition matrices that adapt these layers while keeping the pre-trained weights unchanged. This approach significantly addresses the inference latency introduced by adapters and the fitting challenges of prefix tuning.

In LoRA, the adapter matrices $A$ and $B$ are updated with the same learning rate, but using the same rate for both may not effectively learn the features. LoRA+ extended this method by introducing independent learning rates for matrices $A$ and $B$ with a fixed ratio, improving the method's efficiency. The DoRA Liu et al. (2024b) method combines weight decomposition to achieve learning capabilities similar to full fine-tuning without sacrificing LoRA's inference efficiency. PiSSA optimizes the compact parameter space by representing the matrices in the model as the product of two trainable matrices, augmented with a residual matrix for error correction. Using Singular Value Decomposition (SVD), PiSSA initializes the dominant singular values and vectors to train these matrices, while keeping the residual matrix static during fine-tuning. OLoRA Büyükakyüz (2024) leverages QR decomposition to initialize the adaptation matrices during the fine-tuning process, ensuring that these matrices are orthogonal. This orthogonal initialization helps maintain the stability of the parameter space during optimization. LoRA-GA and PiSSA are similar in form, but they differ in that LoRA-GA initializes $A$ and $B$ by computing the initial gradient, thereby closely approximating full fine-tuning.

## 3 METHOD

### 3.1 MOTIVATION

The primary motivations are as follows:

Firstly, current research on LoRA has reached a saturation point, underscoring the need for a new direction in Parameter-Efficient Fine-Tuning (PEFT) techniques.

Secondly, existing PEFT methods lack a thorough exploration and effective utilization of the original model weights.

### 3.2 BONE: BLOCK-AFFINE UPDATE MATRICES

To design a novel PEFT structure that rivals LoRA, we conducted an in-depth study of dense matrices. Current theories suggest that LLM weights exhibit sparse properties. For example, LoRA utilizes low-rank matrices to update dense matrices, while techniques like GQA Ainslie et al. (2023) and MQA Shazeer (2019) employ shared mechanisms to reduce parameter overhead with remarkable effectiveness.

Inspired by these approaches, we hypothesize that dividing LLM weights into multiple subspaces allows different subspaces to share a single low-rank matrix for updates. As illustrated in the figure 2, we name this structure Block-Affine Adaptation (Bone).

It is evident that the ways to partition LLM weights are diverse, as illustrated in the figure. Different partitioning methods influence the shape of the Bone matrix, the computation logic, and ultimately, the processing speed. In this work, we propose the Best Bone structure. For a pre-trained weight matrix $W \in \mathbb{R}^{d \times k}$, we update it using a single matrix: $W + \Delta W = W + expand(B)$, where $B \in \mathbb{R}^{r \times d}$, and the rank $r \ll k$. This design not only retains the simplicity and ease of use characteristic of LoRA but also achieves higher computational efficiency. The calculation is expressed as follows:

$$Y = Wx + \Delta Wx = Wx + sum(x)B \tag{1}$$

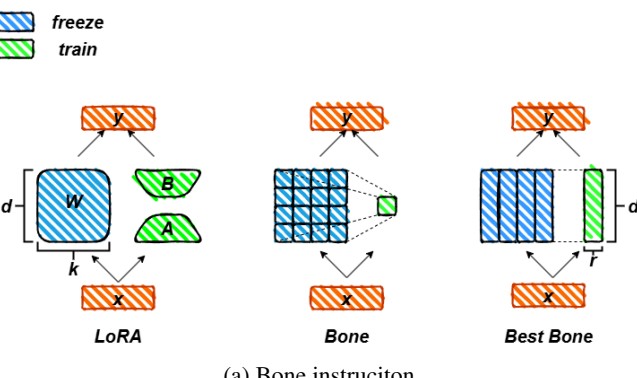

(a) Bone instruciton

Figure 2: Bone requires training only a single low-rank matrix initialized to zero. Best Bone represents the optimal design structure of Bone. Subsequent experiments default to using Best Bone.

### 3.3 "WEIGHT GUIDE"

It is well-known that LLMs have extremely large parameters, with complex interactions among their internal weights, but most LoRA variants utilize original weight information minimally, with only a few leveraging it during initialization to enhance performance. To increase the utilization of this information and promote internal interactions among weights, we propose a novel concept called "Weight Guide" This approach ensures that the trainable matrices are consistently guided and constrained by the original weights throughout every step of the training process. As illustrated in Figure 1b, LoRA variants interact with the original weights only once during initialization to extract essential components. In contrast, the Bone structure is guided (constrained) by the original weights after each update, significantly increasing the new weights' utilization of the original weights. As the training steps increase, the influence of the original weights becomes more pronounced. This is evident from the loss curves, which show that the differences between Bone and other methods grow larger in the later stages of training.

So, how can we extract the critical information from the original weights? A straightforward approach is to apply a learnable matrix directly to the original weights using the Hadamard product. This method essentially assigns a weight to each element, allowing the learnable matrix to autonomously determine the importance of individual elements. Although this method is simple and straightforward, the elements within the weights remain isolated, making it impossible to learn interactions between them. Therefore, we replace the Hadamard product with matrix multiplication, essentially applying a simple linear transformation. This approach links the entire weight matrix space, providing a better chance to discover the optimal solution. The corresponding formula is shown below:

$$\begin{aligned} \Delta W_{n,n} &= W_{n,n} \odot bone_{n,n} \\ \Delta W_{n,n} &= W_{n,n} \otimes bone_{n,n} \end{aligned} \tag{2}$$

### 3.4 BAT: BLOCK-AFFINE TRANSFORMATION

In the previous section, we introduced Bone, which is both simple and efficient. However, applying the same updates to different subspaces of the weight matrix $W \in \mathbb{R}^{d \times k}$ is clearly counterintuitive. To allow for more flexibility in the update directions of different subspaces, we combine Bone with

"Weight Guide", proposing a new method called Block-Affine Transformation (Bat). To balance computational efficiency, we divide the original weights into smaller subspaces compared to Bone. Here, we name the trainable matrix $bone \in \mathbb{R}^{r \times d}$, where its rank $r$ is replaced with block size $b$. The formula for Bone-col is as follows:

$$
\begin{aligned}
W_{k/b,d/b,b,b} &= Reshape(W_{d,k}) \\
bone_{d/b,b,b} &= Reshape(bone_{b,d}) \\
\Delta W_{d,k} &= Reshape(W_{k/b,d/b,b,b} \otimes bone_{d/b,b,b} + bone_{d/b,b,b})
\end{aligned}
\tag{3}
$$

Similar to Bone, Bat can also adopt various grouping methods, as illustrated in the figure 3. In Section 6, we compared the performance of different partitioning strategies for the structure. The formula for Bone-row is as follows:

$$
\begin{aligned}
W_{d/b,k/b,b,b} &= Reshape(W_{d,k}) \\
bone_{k/b,b,b} &= Reshape(bone_{b,k}) \\
\Delta W_{d,k} &= Reshape(W_{d/b,k/b,b,b} \otimes bone_{k/b,b,b} + bone_{k/b,b,b})
\end{aligned}
\tag{4}
$$

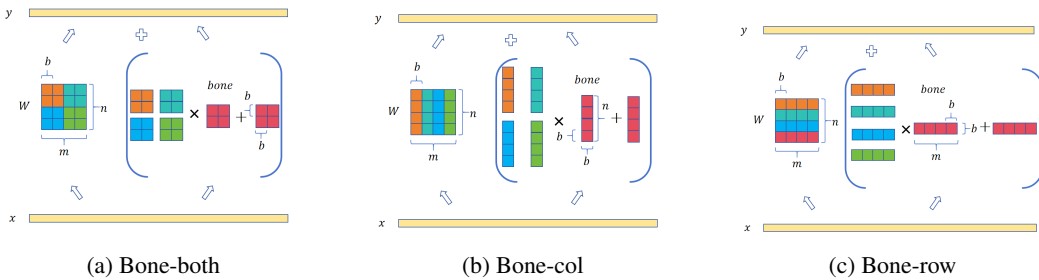

(a) Bone-both     (b) Bone-col     (c) Bone-row

Figure 3: A comparison between the visualizations of LoRA and Bone structures reveals that when $n = m$ in $W$, setting bone_b to $2 \times$ lora_r ensures that the trainable parameters of both structures are equal. The figure illustrates three different grouping methods for the Bone structure. Grouping by rows and columns allows for seamless adaptation to any LLM structure, making it easier to adapt to different LLM architectures.

## 4 EXPERIMENTS

In this section, we evaluate the performance of Bone on various benchmark datasets. Initially, we assess Natural Language Understanding (NLU) capabilities using a subset of the GLUE dataset with the robert-base model. Subsequently, we evaluated the Natural Language Generation (NLG) capabilities by fine-tuning the LLM.

The experiments were conducted on 4×NVIDIA 4090 24G GPUs.

### 4.1 EXPERIMENTS ON NATURAL LANGUAGE UNDERSTANDING

**Models and Datasets**  We fine-tune the RoBERTa-base model on several datasets from the GLUE benchmark, including MNLI, SST-2, CoLA, QNLI, and MRPC. Performance is evaluated on the development set using accuracy as the primary metric.

**Implementation Details**  The experimental hyperparameter settings were aligned with those in the LoRA repository, but training was conducted using a single 4090 GPU. Each experiment is conducted with 3 different random seeds, and the average performance is reported.

**Results**  As shown in Table 1, Bone demonstrates outstanding performance, particularly on the CoLA dataset, where it exhibits significantly faster convergence and superior data-fitting capabilities, far surpassing LoRA and Pissa.

Table 1: The results of fine-tuning RoBERTa-base using Bone and various LoRA variants were compared on a subset of the GLUE benchmark.

| Method | Trainable | MNLI | SST-2 | CoLA | QNLI | MRPC |
|--------|-----------|------|-------|------|------|------|
| LoRA | 0.236% | $85.63_{\pm 0.01}$ | $\mathbf{94.03_{\pm 0.02}}$ | $62.40_{\pm 0.71}$ | $91.37_{\pm 0.97}$ | $87.98_{\pm 0.23}$ |
| Pissa | 0.236% | $\mathbf{85.72_{\pm 0.40}}$ | $93.64_{\pm 0.13}$ | $67.28_{\pm 0.59}$ | $91.40_{\pm 0.54}$ | $88.11_{\pm 0.24}$ |
| Bone | 0.236% | $85.71_{\pm 0.32}$ | $93.60_{\pm 0.07}$ | $\mathbf{72.86_{\pm 3.13}}$ | $\mathbf{91.43_{\pm 0.76}}$ | $\mathbf{88.14_{\pm 0.60}}$ |

## 4.2 EXPERIMENT ON NATURAL LANGUAGE GENERATION

**Models and Datasets**  To verify the generalizability of Bone, we conducted more comprehensive experiments on LLM. we conducted 3 more task finetuning experiments on LLM: *math*, *code*, and *chat*.

**1.** *Math*: We trained our model on a 395k subset of MetaMathQA Yu et al. (2023), a dataset bootstrapped from other math instruction tuning datasets like GSM8K Cobbe et al. (2021) and MATH Yu et al. (2023), with higher complexity and diversity.
**2.** *Code*: We train our model on a 100k subset of CodeFeedback Zheng et al. (2024b), a highquality code instruction dataset, removing explanations after code blocks. The model is tested on HumanEval Chen et al. (2021).
**3.** *Chat*: We train our model on a 70k subset of WizardLM-Evol-Instruct Xu et al. (2024). We test our model on the MT-Bench dataset Zheng et al. (2024a), which consists of 80 multi-turn questions designed to assess LLMs on multiple aspects. We used GPT-4o to judge the quality of responses, as shown in lm-sys/FastChat.

**Implementation Details**  The hyperparameter settings for this experiment were kept equal, while the train steps were adjusted according to the specific fine-tuning datasets used. It is worth noting that the weights of LLaMA2-7B are not fully symmetric, making it impossible to perfectly align the trainable parameters when comparing Bone and LoRA. To address this, we set the rank $r$ of LoRA to 36 and the rank $r$ of Bone to 64, ensuring that Bone uses fewer parameters than LoRA to demonstrate its superiority. Each experiment is conducted with 2 different random seeds, and the average performance is reported.

**Result**  The results, as shown in Table 2 and Figure 1a, demonstrate that Bone outperforms other PEFT methods in terms of convergence speed, data fitting, and generalization capabilities. Bone demonstrates outstanding performance across three different tasks. On LLaMA2-7B, Bone achieves results that surpass Pissa, despite using fewer parameters than LoRA and its variants. On RWKV6-7B, Bone and LoRA have the same number of trainable parameters, yet Bone consistently outperforms LoRA and its variants across all tasks.

## 4.3 EFFECT OF RANK $r$

This subsection explores the upper limits of the Bone structure by varying the rank $r$ in the Bone matrix. Comparative experiments were conducted by fine-tuning LLaMA2-7B on the MetaMathQA dataset and validating on GSM8K and Math benchmarks. The test results, as shown in Table 3, demonstrate that the fine-tuning performance improves as the value of b increases. Notably, when $r = 16$, the Bone structure, with only one-quarter of the trainable parameters compared to PiSSA, surpasses PiSSA's performance on the GSM8k benchmark. However, its performance on the Math benchmark is only 3.73. The GSM8K score surpasses that of PiSSA, but the Math score is significantly lower, indicating The size of $r$ impacts the model's ability to understand unseen data. Based on this observation, we hypothesize that when the rank is too small, it significantly limits the model's generalization ability.

Table 2: We fine-tuned LLMs using Bone and various LoRA variants, and evaluated performance on GSM8k, Math, HumanEval, and MT-Bench.

| Model | Strategy | Trainable | GSM8K | Math | HumanEval | MT-Bench |
|---|---|---|---|---|---|---|
| Llama2-7B | LoRA | 89.9M | 40.75 | 5.22 | 17.68 | 3.73 |
| | OLoRA | 89.9M | 42.93 | 6.51 | 21.12 | 4.03 |
| | PiSSA | 89.9M | 43.89 | 6.92 | 22.25 | 4.11 |
| | Bone | 87.0M | **48.16** | **8.58** | **24.08** | **4.31** |
| RWKV 6-7B | LoRA | 88.1M | 38.13 | 6.06 | - | - |
| | PiSSA | 88.1M | 40.48 | 6.12 | - | - |
| | Bone | 88.1M | **41.73** | **6.52** | - | - |
| Mistral-7B | LoRA | 89.1M | 65.17 | 15.82 | 39.02 | - |
| | PiSSA | 89.1M | **67.01** | 18.13 | 40.85 | - |
| | Bone | 88.1M | 66.94 | **18.85** | **41.76** | - |

Table 3: Comparing different values of rank ($r$)

| Model | rank | Trainable | GSM8K | Math |
|---|---|---|---|---|
| Llama2-7B | 16 | 21.7M | 45.90 | 3.77 |
| | 32 | 43.5M | 46.18 | 7.43 |
| | 64 | 87.0M | 48.16 | 8.58 |
| | 128 | 174.0M | 53.49 | 10.08 |

## 4.4 BONE VS BAT

To validate the effectiveness of the "Weight Guide", we fine-tuned LLaMA2-7B and RWKV6-7B using both Bone and Bat on the MetaMathQA dataset and evaluated their performance on Math and GSM8K. As shown in the table, Bat, equipped with "Weight Guide", achieves significant improvements in performance metrics compared to Bone.

Table 4: Comparing Bone, Bat on math tasks

| Model | Strategy | Trainable | GSM8K | Math |
|---|---|---|---|---|
| Llama2-7B | Bone | 87.0M | 48.16 | 8.58 |
| | Bat | 87.0M | **49.36** | **8.88** |
| RWKV6-7B | Bone | 55.1M | 41.73 | 6.52 |
| | Bat | 55.1M | **42.76** | **6.60** |

## 4.5 ABLATION EXPERIMENTS ON DIFFERENT GROUPING METHODS FOR BAT

In this subsection, we explore the impact of different grouping methods in the Bat structure on model fine-tuning performance. Due to structural differences in the weight matrix W, the Bat-free grouping requires manual configuration, which is inconvenient. Therefore, this subsection only compares row-wise and column-wise grouping, both of which can be easily extended to any structure. We fine-tuned LLaMA2-7B on the MetaMathQA dataset and validated the results on GSM8k and Math. The results are shown in Table 5. Since in LLaMA2-7B, the dimension of gate_proj in the MLP part is $(4096, 11008)$, this leads to an asymmetry between row-wise and column-wise grouping in the Bat structure, making it difficult to align parameter counts. Although Bat-row uses 15M fewer parameters than Bat-col, it still delivers excellent performance. However, this discrepancy in

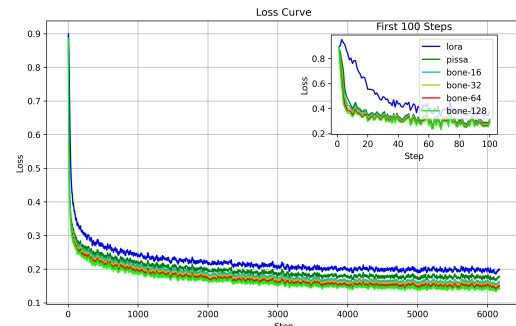

Figure 4: Training loss curves of Bone with different rank $r$ on the MetaMathQA dataset.

parameter counts makes it challenging to accurately evaluate the differences between the grouping methods.

To explore the differences between the two grouping methods and the effect of parameter count, we added a comparative experiment with RWKV6-3B, as RWKV6's symmetrical structure ensures that the trainable parameter count is the same whether using row-wise or column-wise grouping during Bat fine-tuning. This allows for a fairer comparison between Bat-row and Bat-col. The experimental results, shown in Table 5, indicate that the difference between the two is minimal, with both performing well. Therefore, we believe that Bat can effectively fit data regardless of the grouping method used. The key factor influencing Bat's performance remains the block size. However, this doesn't imply that different grouping methods are meaningless. As more LLMs begin to use techniques like GQA and MLA Liu et al. (2024a) to reduce KV cache overhead Dai et al. (2024); Lee et al. (2024); Shazeer (2019), the main weight matrices become smaller, and Bat will need to adjust its grouping or employ other techniques to adapt to these new technologies.

Table 5: Comparing Bat-row, Bat-col on math tasks

| Model | Strategy | Trainable | GSM8K | Math |
|---|---|---|---|---|
| Llama2-7B | Bat-row | 72.8M | 45.76 | 7.82 |
| | Bat-col | 87.0M | **49.36** | **8.88** |
| RWKV6-3B | Bat-row | 55.1M | 25.93 | 3.12 |
| | Bat-col | 55.1M | 25.25 | 3.09 |

### 4.6 ABLATION EXPERIMENTS ON THE SPECIFIC IMPLEMENTATION OF "WEIGHT GUIDE"

Evaluating the best computation method for promoting internal weight feature fusion on RWKV6-3B: As seen in the table 6, Bone-Hadamard is significantly weaker compared to matrix multiplication.

The formula for Bone-Hadamard is as follows:

$$
\begin{aligned}
W_{k/b,d/b,b,b} &= Reshape(W_{d,k}) \\
bone_{d/b,b,b} &= Reshape(bone_{b,d}) \\
\Delta W_{d,k} &= Reshape(W_{k/b,d/b,b,b} \odot bone_{d/b,b,b} + bone_{d/b,b,b})
\end{aligned}
\tag{5}
$$

## 5 RESOURCE AND EFFICIENCY

Table 7 compares the training resources and token throughput required for fine-tuning RWKV6 using LoRA, Bone, and Bat on a single 4090 GPU. The specific fine-tuning settings are as follows: batch size = 1, context length (ctx_len) = 512.

Table 6: Block-Affine on math tasks

| Model | Strategy | Trainable | GSM8K | Math |
|---|---|---|---|---|
| RWKV6-3B | Bat | 55.1M | 25.93 | 3.32 |
| | Bat-Hadamard | 55.1M | 22.44 | 2.67 |

The results show that Bone has the highest computational efficiency, being nearly 10% faster than LoRA while also being more memory-efficient. However, Bat incurs significantly higher memory usage due to large intermediate values and is slower in comparison.

At the end of the table, we provide the actual resource costs for fine-tuning RWKV6 on the Meta-MathQA dataset using 4 NVIDIA 4090 GPUs, with checkpoint techniques applied.

Therefore, a key focus of our future work will be improving the Bat operator to enhance token throughput and reduce memory usage.

Table 7: Resource and efficiency

| Model | Strategy | Trainable | GPU Memory | Token throughput |
|---|---|---|---|---|
| RWKV-3B | LoRA | 55.1M | 12074 MB | 3.62 kt/s |
| | Bone | 55.1M | **11052** MB | **3.99** kt/s |
| | Bat | 55.1M | 22978 MB | 2.16 kt/s |
| RWKV-3B | LoRA(use checkpoint) | 55.1M | 4*15328 MB | 15.6 kt/s |
| | Bone(use checkpoint) | 55.1M | $4*\mathbf{15304}$ MB | **16.0** kt/s |
| | Bat(use checkpoint) | 55.1M | 4*15305 MB | 14.2 kt/s |

# 6 CONCLUSION

This work, inspired by GQA and MQA, leverages the sparsity of LLM weights to design the Block-Affine Adaptation (Bone) structure. In Bone, the original weights are divided into multiple sub-spaces, all of which share a single low-rank matrix initialized to zero for updates. Extensive experiments demonstrate that Bone consistently outperforms LoRA and its variants across various tasks, while also offering superior computational efficiency. To break the limitations of LoRA and effectively utilize the information from the original weights, we propose the "Weight Guide" theory, which enables significant improvements through simple linear transformations.

By integrating these innovations, we introduce a new structure called Block-Affine Transformation (Bat). Bat not only validates the effectiveness of "Weight Guide" but also addresses the limitation of Bone, where identical updates are applied to all subspaces. Experimental results show that Bat surpasses Bone on multiple tasks, although with a trade-off in computational efficiency.

Bone brings new possibilities to existing LLM PEFT techniques. Instead of focusing solely on optimizing LoRA, we should shift our attention to innovative PEFT methods that are better suited to the architecture of LLMs.

# 7 FUTURE WORK

Bone and "Weight Guide" are merely a starting point, offering a foundation for researchers to explore additional branches inspired by Bat. There are many directions for future work.

**1.** Can Bone adapt to multimodal tasks and other complex scenarios?
**2.** Can "Weight Guide" be implemented using alternative computation methods?

We welcome the community to provide additional suggestions and conduct further tests.

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
