# OpenReview forum: "BONE: BLOCK AFFINE TRANSFORMATION AS PARAMETER EFFICIENT FINE-TUNING METHODS FOR LARGE LANGUAGE MODELS"
_ICLR.cc/2025/Conference — ICLR 2025 Conference Withdrawn Submission_

### Official Review · Reviewer_29jE · 2024-10-29

**Soundness:** 2
**Presentation:** 1
**Contribution:** 2
**Rating:** 3
**Confidence:** 4

**Summary:**

This paper proposes "Bone," a Parameter-Efficient Fine-Tuning (PEFT) method for large language models (LLMs).
Bone divides the pre-trained weight matrix W into smaller blocks, applies matrix multiplications with a Bone matrix on these blocks, and then adds the bone blocks $Y_{b} = (W_{b} ∆W + ∆W) X$.
This authors claim this approach will continually integrate information from the pre-trained weights throughout training, and the method thus seeks to "guide" the trainable matrices (Bone matrices) with the original weights to strengthen the internal weight connections in the model and enhance weight utilization efficiency.
The proposed method's effectiveness is evaluated through experiments on two LLM architectures, RWKV6 and LLaMA2, where it reportedly achieves superior performance.

**Strengths:**

- A novel approach to PEFT: The effort to design a new PEFT method, rather than solely optimizing LoRAs, is both innovative and refreshing.
- The authors demonstrate that Bone outperforms LoRA in certain settings.

**Weaknesses:**

- Main weakness - computational overhead: Although Bone and the baseline LoRA(s) are constrained to use an equal number of parameters, Bone requires more computational power due to its multiple block-wise matrix multiplications. Consequently, using the total number of iterations as a measure of convergence speed is less meaningful and it is unclear how Bone compares to LoRA in terms of convergence speed. A plot of convergence versus wall clock time would provide a clearer view of efficiency. Additionally, the increased memory usage is notable, and while checkpointing may reduce it, this approach is orthogonal to the PEFT method itself, making any efficiency gains from checkpointing less relevant.
- Limited evaluation tasks: The evaluation is restricted to a small set of tasks, limiting the ability to fully assess Bone's efficacy. Expanding the range of tasks to the task set in PiSSA [1] would provide a more robust evaluation.
- Lack of discussion on result variance: Only single values are reported, with no mention of result variance. Given the small observed improvements, experiments with varying random seeds and reporting mean and standard deviation would help readers better assess the method’s reliability and effectiveness.

[1]PiSSA: Principal Singular Values and Singular Vectors Adaptation of Large Language Models, Meng et al, NeurIPS 2024.

**Questions:**

The authors state that their method guides (or constrains) the trainable Bone weights using pre-trained weights. However, it is unclear to the reviewer how matrix multiplication accomplishes this. Could the authors clarify this mechanism?

---

> ### Author Response · Authors · 2024-11-14
>
> Thank you very much for your review. Below, I will respond to your questions and provide explanations for the "Weaknesses" section, along with feasible solutions.
>
> **Questions Section:**
>
> The inspiration for the "Weight Guide" comes from the statement in the Pissa paper regarding the use of SVD decomposition to extract important components from the original weights, as well as attention mechanisms. The results in Pissa demonstrate that the original weights can indeed be utilized. Therefore, we used the trainable matrix "Bone" to allow it to autonomously determine which elements are useful and which are not. So, when we use Hadamard operations, it essentially adds corresponding weights to each element (but the training matrix "Bone" fully controls this). At the same time, we hypothesize that different elements may also influence each other, which is why we replaced the Hadamard operation with matrix multiplication. This way, we can aggregate the entire rows and columns of elements together.
>
> In Section 4.5, we conducted ablation experiments to compare the performance of Hadamard versus matrix multiplication. To more intuitively show the performance improvement brought by the "Weight Guide," we added new ablation experiments comparing the performance without the "Weight Guide."
>
> **Weaknesses Section:**
>
> 1. Recently, we modified the Bone structure and computational logic, making Bone faster than Lora, while the performance metrics still outperform Lora(s).
>
>     Latest test results on RWKV6-3B with the same configuration:
>
>     - **Bone-new**    [Trainable: 55.1M, GPU Memory: 11052MB, Token throughput: 3.95kt/s]
>     - **Lora (Pissa)** [Trainable: 55.1M, GPU Memory: 12074MB, Token throughput: 3.62kt/s]
>     - **Bone-old**      [Trainable: 55.1M, GPU Memory: 22978MB, Token throughput: 2.16kt/s]
>
>     Evaluation results on LLaMA2-7B:
>
>     - **Bone-new**[Trainable: 87.0M, GSM8K: 47.67, Math: 8.54]
>     - **Pissa**       [Trainable: 89.9M, GSM8K: 43.52, Math: 6.92]
>     - **Bone-old** [Trainable: 87.0M, GSM8K: 49.36, Math: 8.88]
>
>     Of course, we will also provide a plot of convergence versus wall clock time.
>
> 2. We will include more comprehensive evaluation tasks, and we are also planning to add tests in the vision domain.
> 3. The training framework code for Bone was migrated from the Pissa repository and tested using open-source testing libraries. In order to reduce randomness, both training and testing were repeated more than twice. However, due to minimal differences, we did not average the results. In future improvements, we will add more details and experiments to ensure the fairness of the experiments.
>
> Finally, we greatly appreciate your valuable suggestions and feedback. We hope that the explanations and improvements mentioned above will resolve your concerns and increase your recognition of this paper.

---

> > ### Comment · Reviewer_29jE · 2024-11-23
> >
> > Thank you for your detailed response and for addressing the points raised in my review. While I remain unconvinced, I appreciate your explanation of the Weight Guide and the efforts to improve the Bone structure. However, I note that the paper is still in its original form and has not been revised. I look forward to seeing:
> >
> > 1. The wall-clock time speed-up data and associated plot you mentioned.
> > 2. A clearer explanation of what specifically has changed in the "Bone" structure to improve its performance.
> > 3. New evaluation results to further substantiate the claims.
> >
> > This is an interesting and promising piece of work, but it remains a work in progress. My current score reflects the paper’s present state, and I encourage you to incorporate these updates into a future version.

---

> > > ### Author Response · Authors · 2024-11-25
> > >
> > > Thank you for your review. I have submitted a new version of the paper and addressed most of the issues. I hope you can review it again.

---

### Official Review · Reviewer_gSGJ · 2024-11-01

**Soundness:** 3
**Presentation:** 2
**Contribution:** 2
**Rating:** 5
**Confidence:** 4

**Summary:**

The authors propose a PEFT method called BONE that makes use of a weight update matrix that is multiplied by the original weight matrix, and they also make the update matrix block-structured to make it parameter-efficient. Experimental results show the method is more effective than LoRA and PiSSA on several math and reasoning tasks. However, there are limited explanations or explorations on why the method works.

**Strengths:**

- The algorithm is simple to implement. The main idea is to multiply the weight update with the original weight matrix, and divide the weight update with shared blocks to reduce the number of updatable parameters.

- Evaluations on RWKV6 and Llama on several math and reasoning tasks show the effectiveness of the method, beating competing methods like LoRA and PiSSA by a fairly large margin.

- There are several ablation studies comparing multiplying the weight update with the original matrix using the Hadamard product, and also alternative decomposition of the weight update matrix using rows and columns.

**Weaknesses:**

- The authors propose the concept of "weight guide" to explain their approach but I found it difficult to understand. The main difference compared to LoRA is the multiplication of the weight update \Delta W with the original weight matrix W. But why does this form of update help compared to LoRA? The authors claim this approach "ensures that the trainable matrices are consistently guided and constrained by the original weights throughout every step of the training process". This is still rather vague and the authors should offer more in-depth explanation of why their approach works better.

- Some of the results presented in the tables do not seem to be consistent with the literature. For example, when we compare Table 3 of the paper with Figure 3d of the PiSSA paper, when the accuracy of MATH is around 6.92 for PiSSA (rank=64), the corresponding accuracy for LoRA should be between 5-6, not as low like 0.44. Please check the results.

**Questions:**

Please address the questions listed in the weaknesses section above.

---

> ### Author Response · Authors · 2024-11-25
>
> Regarding the issue you mentioned about the Lora metric of 0.44 in Table 3, I would like to clarify that this data is not incorrect. I retrained the model five times, and the results were consistently around 0.44. This is likely related to my Lora settings. Therefore, in the latest training, I adjusted the Lora settings, and now its performance appears to be normal.
>
> Thank you for your review. I have submitted a new version of the paper and addressed most of the issues. I hope you can review it again.

---

### Official Review · Reviewer_E5G2 · 2024-11-03

**Soundness:** 3
**Presentation:** 3
**Contribution:** 3
**Rating:** 5
**Confidence:** 4

**Summary:**

The paper proposes BONE (Block Affine Transformation), a novel parameter-efficient fine-tuning (PEFT) method aimed at improving large language model (LLM) training without the high memory overhead associated with full-parameter tuning. Unlike previous approaches, such as Low-Rank Adaptation (LoRA) and its variants, BONE introduces a "Weight Guide" mechanism, where the original weights guide the training matrices in each update, ensuring better weight utilization and faster convergence. This framework builds on the Block Affine transformation method, where matrix operations are structured in groups to preserve essential weight interactions and minimize computation.

**Strengths:**

1.Block Affine Transformation: The BONE framework applies block-wise transformations to optimize memory and computational efficiency, promoting better internal feature fusion.
2.Experimental Validation: Extensive tests on models such as LLaMA2 and RWKV6 show that BONE outperforms other LoRA variants in terms of convergence speed, data fitting, and generalization on tasks like GSM8K, MetaMathQA, HumanEval, and MT-Bench.

**Weaknesses:**

1.The Writing is a Little Poor: There are typos in the writing. There should be punctuation after the equations, and the table should have a more beautiful structure.
2.Generality for Other Modalities: The current study focuses on language tasks, and its application to other data modalities remains unexplored.

**Questions:**

1. Generality for Other Modalities: What about the model performance on the vision tasks?

2. Memory Usage: In Table 7, does the LoRA also use the checkpointing method to reduce memory? If not, you should compare LoRA and BONE in the same setting.

---

> ### Author Response · Authors · 2024-11-14
>
> Thank you for your recognition of this paper. Since this is my first time writing a paper, I acknowledge that there are areas where the writing, formula arrangement, and table structure could be improved. I will actively address the weaknesses you pointed out. Additionally, I am adding more experimental content, such as testing T5 on GLUE and vision tasks.
>
> Below, I will answer the questions you raised:
>
> 1. We are currently adding vision tasks-related experiments.
> 2. The Bone method has been updated to a more efficient version, so we will revise this section to make it easier for readers to review and compare.
>
> Once again, thank you for your valuable suggestions. I will continue to update the paper.

---

> ### Author Response · Authors · 2024-11-25
>
> Thank you for your review. I have submitted a new version of the paper and addressed most of the issues. I hope you can review it again.

---

### Official Review · Reviewer_WTtk · 2024-11-03

**Soundness:** 1
**Presentation:** 1
**Contribution:** 2
**Rating:** 3
**Confidence:** 5

**Summary:**

This paper suggests using block affine transformations for fine-tuning models.

The proposed method was examined on language modelling tasks using two language models.

In the experimental analyses, the proposed Bone outperforms the baseline PiSSa.

**Strengths:**

The proposed idea is interesting.

In the experimental analyses, the proposed Bone outperforms the baseline PiSSa.

**Weaknesses:**

Although the proposed idea is interesting, there are several weaknesses in the paper.

First, there are several over-claims. For instance, the authors claim that "To address these issues, we introduce a novel theory, “Weight Guide” aimed at continuously guiding trainable matrices through the original weights during training to enhance the utilization of weight information." However, there are not theoretical results elucidating the proposed Bone or the framework.

Second, the explanation of the proposed Bone should be improved. For instance, it is not clear how full-rank and low-rank bone matrices are constructed and employed during training and inference.

Finally, the experimental analyses should be improved by examining the proposed Bone in comparison with the other LoRA methods using different language and vision models in various different tasks.

**Questions:**

How does the proposed Bone perform in comparison with the other LoRA methods using different language and vision models in various different tasks?

How do the results (accuracy, convergence rate, memory consumption etc.) change for different ranks?

---

> ### Author Response · Authors · 2024-11-25
>
> Thank you for your review. I have submitted a new version of the paper and addressed most of the issues. I hope you can review it again.

---

### Official Review · Reviewer_mY5r · 2024-11-03

**Soundness:** 3
**Presentation:** 2
**Contribution:** 3
**Rating:** 6
**Confidence:** 4

**Summary:**

The paper titled **"BONE: Block Affine Transformation as Parameter-Efficient Fine-Tuning Methods for Large Language Models"** introduces an innovative parameter-efficient fine-tuning (PEFT) method called **Bone (Block Affine)**. This method aims to improve the adaptation efficiency of large language models (LLMs) while overcoming the limitations of existing approaches, such as LoRA (Low-Rank Adaptation) and its variants. The authors propose a new concept termed **"Weight Guide"**, which ensures the continuous and effective utilization of original weight information throughout training. The Bone structure utilizes block affine operations, enhancing internal interactions among weights, thereby facilitating faster convergence and superior data fitting. Extensive experiments across LLaMA2 and RWKV6 architectures reveal that Bone outperforms existing PEFT techniques, achieving better performance metrics on tasks such as MetaMathQA, GSM8K, and HumanEval, without the complexity of additional initialization steps or overhead.

**Strengths:**

1. **Originality**
   The paper presents a significant advancement in the domain of Parameter-Efficient Fine-Tuning (PEFT) methods for large language models (LLMs) through the introduction of the **Bone (Block Affine)** structure. The proposed approach innovatively addresses limitations in existing methods like LoRA by introducing the concept of **"Weight Guide"**. This mechanism ensures continuous interaction with the original model weights during training, unlike previous methods that primarily focus on initialization. The originality here is twofold: the method provides a more efficient way to harness original weight information, and it formulates a novel structural approach to enhance convergence and model performance.

2. **Quality**
   The research is methodologically sound and thoroughly evaluated. The authors conduct comprehensive experiments across prominent LLM architectures, such as **LLaMA2** and **RWKV6**, and under various parameter configurations. The results demonstrate that Bone consistently outperforms established methods like PiSSA and LoRA, showcasing improvements in convergence speed and data fitting. Moreover, the paper includes detailed **ablation studies** to isolate the effects of specific design choices, such as the use of matrix multiplication over the Hadamard product and the impact of different grouping methods. The rigorous experimentation lends substantial credibility to the claims made.

3. **Clarity**
   The paper is well-structured and generally easy to follow, which is crucial for ICLR's broad audience of machine learning researchers and practitioners. The authors clearly articulate the motivation behind their work and explain the technical details of the Bone method step-by-step. Figures and tables are effectively used to illustrate complex ideas and results, such as loss curves and the architecture of the Bone structure. However, some areas, particularly those involving mathematical derivations and complex architectural nuances, could benefit from more accessible explanations to ensure wider comprehension among readers who may not be experts in PEFT techniques.

4. **Significance**
   The proposed Bone method has substantial implications for the field of large-scale model fine-tuning. As LLMs are widely used in academia and industry, methods that can achieve efficient and effective fine-tuning without significant computational overhead are highly impactful. The paper demonstrates that Bone not only accelerates convergence but also enhances generalization, making it a highly valuable contribution. The significance is further underscored by the adaptability of the method across different architectures, positioning it as a potentially transformative approach for future research and practical applications in LLM fine-tuning.

**Weaknesses:**

1. **Memory Efficiency Trade-offs**
   - The paper discusses Bone's higher memory requirements compared to LoRA and its variants in Section 5, where the authors provide a table detailing GPU memory usage and token throughput. Specifically, the results in Table 7 indicate that Bone’s memory usage is significantly higher, even when checkpointing strategies are used. Given that one of the primary goals of PEFT methods is to minimize resource demands, this is a major drawback. The authors could improve this aspect by suggesting or exploring more efficient memory management techniques, such as memory-efficient matrix operations or gradient checkpointing optimizations that specifically target Bone's unique structure.

2. **Lack of Full Fine-Tuning Comparisons**
   - The absence of a direct comparison with full fine-tuning methods weakens the argument that Bone can serve as a viable substitute. Although the authors highlight Bone’s performance gains in Sections 4.1 and 4.2, the paper would benefit from an experiment or analysis that directly compares Bone to full-scale fine-tuning on at least one of the datasets used (e.g., GSM8K or MetaMathQA). This addition would provide a clearer benchmark for readers to assess Bone’s true efficiency and performance benefits.

3. **Limited Analysis of Model Generalization**
   - The experimental results focus heavily on specific tasks, such as mathematical reasoning (MetaMathQA and GSM8K) and code evaluation (HumanEval). In Section 4.2, the paper demonstrates impressive performance gains on these benchmarks but does not explore how Bone performs on a broader range of NLP tasks, such as natural language inference or question answering. Including additional datasets, or at least discussing Bone’s expected performance in these other domains, would help to establish the generalizability of the method and make it more compelling for a wider audience.

4. **Complexity of Implementation**
   - The implementation of Bone, as described in Section 3, involves configuring block sizes, managing groupings, and ensuring that the Weight Guide mechanism functions correctly. While the authors claim in the discussion following Figure 2 that Bone simplifies initialization compared to some LoRA variants, the complexity of handling multiple architectural adjustments and grouping strategies could be a barrier to adoption. Providing pseudocode, or releasing an open-source implementation, could alleviate this complexity and make Bone more practical for researchers and practitioners.

5. **Insufficient Theoretical Justification for Design Choices**
   - The choice of matrix multiplication over the Hadamard product for feature fusion, discussed in Section 3.5, is primarily supported by empirical results rather than a thorough theoretical justification. The authors could enhance the paper by elaborating on why matrix multiplication is expected to be superior from a mathematical perspective, potentially referencing work on weight feature interactions in neural networks. Additionally, the justification for the continuous guidance of weights in the "Weight Guide" concept could be deepened with a theoretical analysis to solidify its foundational reasoning.

6. **Ablation Studies Missing Critical Variants**
   - While Section 4.5 presents some ablation experiments, it does not explore all potential variations of the Bone structure. For example, the authors compare Bone with and without the Weight Guide mechanism but do not consider intermediate variants, such as partially constrained guidance or adaptive guidance based on training progress. Furthermore, the impact of different block sizes is discussed, but variations in grouping strategies are only partially explored. Including these additional experiments would offer a more complete understanding of Bone's behavior under different configurations.

7. **Writing Logic**
   - Logical Sequence in the Introduction: The Introduction section, particularly the way the authors summarize their contributions at the end, suffers from a logical inconsistency. As you pointed out, the authors first introduce the concept of "Weight Guide" and only afterward describe the Bone framework, even though Weight Guide is conceptually dependent on the Bone architecture. This sequence could confuse readers, making it appear as though Weight Guide is the central innovation and Bone is a supporting component, which misrepresents the actual relationship between these elements. The Bone structure is the foundational framework, and Weight Guide is a mechanism developed within this framework to enhance its performance.

**Questions:**

- **Address Memory Usage**: Section 5 discusses memory efficiency, but it needs actionable solutions. The authors could explore integrating memory-saving techniques, like more advanced gradient checkpointing tailored to Bone's architecture.
- **Expand Generalization Tests**: Broaden Section 4.2 to include results on diverse NLP tasks, or provide a detailed discussion about the generalizability of Bone to different domains beyond math and code.
- **Simplify Implementation**: Section 3 could be supplemented with pseudocode or a public repository link to facilitate implementation and make Bone more accessible.
- **Strengthen Theoretical Underpinnings**: Provide more rigorous theoretical justifications in Section 3.5 for the choice of matrix multiplication and the Weight Guide mechanism, possibly including mathematical analysis or references to established theories.
- **Broaden Ablation Studies**: Extend Section 4.5 to include more ablation experiments, testing intermediate guidance strategies and more diverse block size configurations.
- **Reorganize the Introduction**: The authors should revise the Introduction to ensure a coherent logical flow. Specifically, they should first clearly explain the motivation behind developing the Bone structure, describing the limitations of existing PEFT methods like LoRA and its variants. Once the Bone framework is introduced as a solution to these issues, the Weight Guide mechanism should be presented as an enhancement that leverages Bone’s unique structure to improve convergence and performance. Also, rearrange your figures and tables to make the paper more well-organized.

---

> ### Author Response · Authors · 2024-11-22
>
> First of all, thank you very much for your detailed suggestions; they have been incredibly helpful to me. Following your advice, I have restructured my paper and made significant revisions, making it much closer to a professional academic article. I am currently adding some experiments and polishing the writing. I believe the new version of the paper addresses 80% of its previous shortcomings. However, I am unsure whether such major revisions comply with the guidelines.
>
> The code for this work is open-source and reproducible, but I did not include the repository in the paper out of concern for violating the double-blind review policy.

---

> ### Author Response · Authors · 2024-11-25
>
> Thank you for your review. I have submitted a new version of the paper and addressed most of the issues. I hope you can review it again.

---

### Note · Authors · 2024-12-07

I have read and agree with the venue's withdrawal policy on behalf of myself and my co-authors.